# Analysis of Body Perception, Preworkout Meal Habits and Bone Resorption in Child Gymnasts

**DOI:** 10.3390/ijerph18042184

**Published:** 2021-02-23

**Authors:** Alessandra Amato, Patrizia Proia, Gaetano Felice Caldara, Angelina Alongi, Vincenzo Ferrantelli, Sara Baldassano

**Affiliations:** 1Department of Psychological, Pedagogical and Educational Sciences, Sport and Exercise Sciences Research Unit, University of Palermo, 90128 Palermo, Italy; alessandra.amato02@unipa.it (A.A.); patrizia.proia@unipa.it (P.P.); 2Istituto Zooprofilattico Sperimentale della Sicilia, 90129 Palermo, Italy; caldarag@tiscali.it (G.F.C.); alongi.angela@gmail.com (A.A.); vincenzo.ferrantelli@izssicilia.it (V.F.); 3Department of Sciences for Health Promotion and Mother and Child Care “G. D’Alessandro”, University of Palermo, 90127 Palermo, Italy; 4Department of Biological, Chemical and Pharmaceutical Sciences and Technologies (STEBICEF), University of Palermo, 90128 Palermo, Italy

**Keywords:** body image perception, children, aesthetic sports, pre-workout eating habits, bone resorption, physical activity

## Abstract

The beneficial effects of physical activity on body image perception and bone are debated among artistic gymnasts. Gymnasts seem to be at greater risk of developing body dissatisfaction, eating disorders and osteoporosis due to inadequate nutrition and attention to the appearance of the body. The objective of this work was to investigate the association between the artistic gymnast and a more favorable body image compared to their sedentary peers and if a preworkout high-carbohydrate meal (HCM; 300 kcal, 88% carbohydrates, 9% protein, 3% fat) or high-protein meal (HPM; 300 kcal, 55% carbohydrates, 31% protein, 13% fat) is able to attenuate bone resorption in young rhythmic gymnasts. Twenty-eight preadolescent female gymnasts were examined. Self-esteem tests were used to analyze body image perception. Preworkout eating habits were examined by short food frequency questions (FFQ) validated for children. The biomarker of the bone resorption C-terminal telopeptide region of collagen type 1 (CTX) was measured in the urine (fasting, postmeal and postworkout). Gymnasts reported higher satisfaction with their body appearance compared to sedentary peers. Of the gymnasts, 30% did not have a preworkout meal regularly, and the timing of the consumption was variable. Bone resorption was decreased by the HCM, consumed 90 min before the training, with respect to the HPM. The study suggests that playing artistic gymnastics is associated with a positive body self-perception in a child. The variability in preworkout meal frequency and timing need attention to prevent inadequate eating habits in light of the ability of the HCM to reduce acute bone resorption.

## 1. Introduction

Preadolescence represents a period of growth and the development of body and personal identity. These changes, which lead to adolescence, especially if rapid, can affect brain health such as self-esteem, body image perception and bone homeostasis. Physical activity helps to increase body self-acceptance [1] and bone growth in children [2]. However, aesthetic sports, such as artistic gymnastics, are considered risk factors for the development of distorted body image, eating disorders and osteoporosis [3,4]. In fact, athletes’ appearance is judged in addition to their performance, and the physical appearance of gymnasts could be the result of a combination of intense physical activity and undernutrition, which could lead to a lean, almost anorexic physique [5]. The problem is that more than fifty percent of total bone mineral content is obtained during preadolescence, which represents the time to achieve an optimal bone mass [6]. Excessive energy expenditure and inadequate nutrition in children with intensive training programs may affect body self-acceptance [7], modify growth during puberty [8], promote bone injuries and predispose children to osteoporosis in older age [9]. The latter, because osteopenia and osteoporosis during adulthood, seem to be related to the degree of bone mineralization during childhood and adolescence [10]. Adequate physical activity [11] and dietary habits with the supply of health-promoting substances is the basis of a healthy diet [12,13]. Therefore, correct dietary habits could be useful to maintain body weight and consequently improve body self-acceptance. However, weight-category sports and aesthetic sports, such as gymnastics or dancing, are associated with a greater risk of developing disordered eating [14,15,16]. A positive body image seems to be linked with more favorable eating attitudes and behaviors [17]. Men usually have greater body appreciation compared to women [18,19]. However, especially in weight-sensitive sports, in which body weight has a high impact on performance [20], understanding the influence of a positive body image in the prevention of disordered eating in depth is a central issue. The few studies conducted on athletes have reported a more favorable body image of athletes compared to nonathletes [21,22,23], and this seems also to improve confidence in sports and the state of flow in physical activity, leading to more successful sports performance [22]. Moreover, less attention has been paid to the associations between a positive body image and physical activity in adolescents [24]. In general, what is known is that participation in physical activity seems to be associated with a more favorable body image [25,26]. Sportive adolescents have greater body image perception compared to nonexercisers [27,28]. It has also recently been shown that participants of leisure and competitive sports reported greater body appreciation, self-esteem and lower body dissatisfaction compared to nonparticipants [29,30,31]. Self-esteem is considered a multidimensional concept that includes self-esteem domains such as school, work, social and emotional spheres and family [32,33]. In fact, there is a close and reciprocal relationship between self-esteem and body image observed starting from primary school children [34] that is also confirmed in adults [35,36]. Thereafter, a positive body image is associated with greater self-esteem and proactive coping [37]. The self-evaluations of physical appearance contribute in a significant way to global self-esteem. Therefore, positive perceptions of body acceptance can be generalized in global self-esteem improvements [38]. Thus, it is possible to evaluate the self-esteem of subjects through their body image perception [33]. It is essential for children to have an adequate preworkout meal because nutrient intake induces acute suppression of bone resorption within two hours following the meal [39]. The international society for clinical densitometry suggests the use of bone turnover markers to monitor acute and chronic bone adaptation in children [40]. The C-terminal telopeptide region of collagen type 1 (CTX) is a very sensitive marker of bone resorption [41]. Unfortunately, data on these biomarkers are very limited in healthy preadolescent children. In particular, information on their change in response to variation of nutrients composition, following physical activity, in preadolescents, is limited. Therefore, the objective of the study was to investigate the association between artistic gymnastics and a more favorable body image of young gymnasts compared to sedentary peers and if a high-carbohydrate (HCM) or high-protein (HPM) preworkout meal is able to attenuate bone resorption in preadolescent rhythmic gymnasts. Moreover, the variables inherently related to body image, such as preworkout eating habits (type of meal consumed, the frequency of consumption and the timing of consumption), were investigated. Based on the previous finding, it was supposed that gymnasts would be associated with more favorable body image perception. They would also have attenuation in bone resorption related to the macronutrient composition of the preworkout meal.

## 2. Materials and Methods

### 2.1. Study Population, Participant Characteristics and Meals

Twenty-eight prepubertal girls participated in the study. The children’s inclusion in the study was based upon the following criteria: (1) taking part in artistic gymnastics at a precompetitive level for at least four years and for 12 h/week; (2) individuals were at the premenarcheal period; (3) individuals were Caucasian females aged 9–12 years old; (4) individuals were clinically healthy; (5) individuals were not receiving medication known to affect bone metabolism; (6) individuals had no history of immobilizing surgery or fractures; (7) individuals were on a maturity level of 1 or 2 on the tanner scale. The participants were not involved in other sports or recreational activities (e.g., dancing, biking) organized by the local community at the time of the study. To verify that they were clinically healthy, the subjects provided a health certificate for competitive sport activity. The certificate is released by their physician following a medical examination, urine test (urinalyses), electrocardiogram at rest and under stress and spirograph (diagnostic test under Italian law to be able to practice competitive sports activities (Ministerial Decree 18/02/1982)). Candidates for the study were excluded if they had a chronic illness and/or if they were taking medications, vitamins or mineral supplements. The parents/guardians of forty-two girls were approached initially, and thirty-eight consented to participate in the study. Ten girls were excluded because they did not meet the inclusion criteria. As a control group (C), for the analysis of body perception, we used a sample of 28 schoolgirls (average age = 10.62 ± 0.7 years) attending schools in the same region (Palermo districts). The inclusion criteria for the control group were: (1) being female; (2) being within the same age range of gymnasts; (3) having completed all the questions measurements; (4) not practicing any sport. The exclusion criteria were the same as above. The gymnast group was divided into two groups, with 14 girls assigned to the high-protein meal (HPM; 300 kcal, 55% carbohydrates, 31% protein, 13% fat) group and 14 assigned to the high-carbohydrate meal (HCM; 300 kcal, 88% carbohydrates, 9% protein, 3% fat) group. The HPM consisted of bread, baked ham and orange juice, and the HCM consisted of cereals and orange juice. Participants were provided with a 500 mL bottle of water to consume ad libitum. In this way, the amount of water introduced by the subjects was fixed. The energy and nutrient content of the meals are described in Table 1. The gymnast (both the HPM and HCM groups) and control groups’ characteristics are presented in Table 2.

The groups did not differ for age, weight, height, BMI, body composition or training characteristics. Specifically, the years of training were 5.4 ± 1.7 for the HPM group and 4.1 ± 1.1 for the HCM group, and both groups performed artistic gymnastics 12 h/week.

### 2.2. Ethics

The study was approved by Ethics Committee Palermo 1, Policlinico Giaccone Hospital, Palermo, Italy (Decision Number: 2/2020–19/02/2020) and was conducted in accordance with the Declaration of Helsinki. Informed written consent was obtained from the parents or legal guardians of each child, and each child gave verbal consent to participate in the study. The parents/guardians provided a full health history questionnaire, and the girls underwent a thorough physical examination and a structured interview.

### 2.3. Multidimensional Test of Self-Esteem for Body Image Perception

To evaluate body image perception, a subscale of the multidimensional test of self-esteem [33] was used. The subscale evaluates how a child feels about themselves regarding their body appearance (e.g., size, hair, skin, etc.) and consists of 25 Likert-type items, with positive and negative statements. The test was administered to the control group and the gymnast group. In the analysis of the multidimensional test of self-esteem, the control group was compared to the gymnast group (which included the gymnasts of both HPM and the HCM groups). Participants were asked to rate their agreement or disagreement with statements on a 4-point scale using anchors of absolutely true and absolutely false. The score assigned was from 4 to 1 point in the positive statements, from “absolutely true” to “absolutely false,” and from 4 to 1 point in the negative statements, from “absolutely false” to “absolutely true.” Higher scores indicated a higher level of body esteem.

### 2.4. Short Food Frequency Questionnaire

The participants were interviewed in the presence of their parents using validated questionnaires designed to report food frequency in children [42,43] adapted to pre-exercise eating habits. The FFQ contained questions about each of the following food items or food groups: fruit, potato chips, homemade cake, fruit drink, milk, yogurt, orange juice, savory snacks and sweet snacks. The frequency scales used were never, once a week, 2–3 times per week and always. One question referred to the time of consumption of the meal before the training section (90, 60, 30 min or just before physical activity). The question about the habit of eating a pre-exercise meal required a yes or no answer.

### 2.5. Anthropometric Measurement

Body weight and body composition (lean mass, fat mass, bone mineral density) were measured after an overnight fast on an electronic scale (Gima 27088; Gima, Italy) calibrated to the nearest 0.1 kg. Barefoot standing height was measured to the nearest 0.1 cm by using a wall-mounted stadiometer (Gima 27335; Gima, Italy). The coefficients of variation for the serial measurements of weight and height were 0.95 and 0.98, respectively. Body mass index (BMI) was calculated as weight, in kilograms, by squared standing height, in meters.

### 2.6. Bone Resorption Experimental Protocol

To avoid interference due to the circadian rhythm of excretion of the markers, the samples were collected at 8:00 in the morning [44,45]. A flow chart of the experimental protocol is described miniaturized in Figure 1A. After overnight fasting, at 0 min, the first urine sample was collected. Then, the gymnasts consumed the assigned pre-exercise meal. Ninety minutes after the meal, before starting the training section, the second sample of urine was collected. Therefore, a sport session competition was simulated. At 180 min, after the end of the session, the third sample of urine was collected. At 240 min, 60 min after the end of the training session, the last sample of urine was collected. Samples were immediately frozen and kept at −80 °C until assayed.

### 2.7. Biochemical Measurements

Human cross-linked C-terminal telopeptides of type I collagen (ß-CTX) were measured by an enzyme-linked immunosorbent assay (Catalog Number: EH3989, Fine-Biotech), as previously reported [46]. Assay values were corrected for urinary dilution by urinary creatinine concentration. The intra- and interassay coefficients of variation were <8% and <10%, respectively. All samples were analyzed in duplicates in the same assay to prevent interassay variation. Urinary creatinine was determined by a standard colorimetric method (Catalog Number: MAK080, Sigma-Aldrich, St. Louis, MO, USA).

### 2.8. Statistical Analysis

We calculated that a minimum of eight participants would be necessary to detect a difference of 20% in CTX with an SD of 12%, a power of 90%, a two-sided significance level of 5% and an effect size of 80%, according to previous studies [47,48,49]. The CTX results are expressed as a percentage of the fasting level. Differences at time points between groups were analyzed by two-way repeated-measures ANOVA with a Sidak test for multiple comparisons. Areas under the curve (AUCs) were calculated with y = 100% as the baseline. The body image perception results were analyzed by the Student’s t-test. Calculations and graphs were all made in GraphPad Prism 6 (GraphPad Software). Data are shown as averages and SEM. Differences resulting in a value of *p* < 0.05 were considered statistically significant.

## 3. Results

### 3.1. Self-Esteem of Body Image

The investigation regarded whether artistic gymnastics, performed at a precompetitive level, influenced body image perceptions compared with sedentary peers. The data analysis revealed significant differences in body self-esteem between the control group and gymnasts. In particular, the gymnast group showed higher body esteem than the control group (Figure 2).

### 3.2. Pre-Exercise Eating Habits

The frequency of food consumption among the young gymnasts was investigated before the training section. Of the young gymnasts, 68% (*n* = 19) regularly consumed a pre-exercise meal before physical activity, and 25% (*n* = 7) of the girls had a pre-exercise meal 2–3 times/week. Only two gymnasts (7%) did not have any pre-exercise meal.

The timing of meal consumption before workout was also examined. Of the children, 39% (*n* = 10) consumed the meal ninety minutes before physical activity, 34% (*n* = 9) of the children had a meal thirty minutes before the training session and 25% (*n* = 6) of the gymnasts consumed the food sixty minutes before the training section. Only one girl (2%) had the meal just before the beginning of the workout. The gymnasts usually prefer to eat fruit (*n* = 18; 64.3%), drink fruit juice (*n* = 20; 71.4%) and eat sweet snacks (*n* = 18; 64.3%); on the other hand, they preferred a milk-based meal or savory snack at least 2–3 times per week before physical activity (Table 3).

### 3.3. Effects of the Pre-Exercise Meal on Bone Resorption

The bone resorption marker CTX was elevated during fasting, and it was reduced after the consumption of the meals in both groups (Figure 1A,B). In particular, the effect of the HPM on bone resorption was transient, with a maximal drop in the CTX levels from the baseline at 90 min after the meal followed by a rise in CTX levels at 180 and 240 min (Figure 1A). The ingestion of the HCM also induced a maximal reduction in CTX levels at 90 min after the meal. The reduction of the bone resorption marker CTX was more lasting. The effect persisted at the end of the training section and 60 min postexercise (Figure 1A), and it was significantly different in comparison with the HPM, as shown also by the AUC at 0–240 min (Figure 1B).

## 4. Discussion

Sports have a positive influence on children’s perception of body image [50,51]. Physical activity may contribute to the formation of children’s personality, to their proper perception, self-esteem and psychological and physical well-being [7,52]. However, so far, for children engaged in aesthetic sports, it is unclear. In fact, they may have a different ideal body image because they are subjected to stronger pressure in regard to their body appearance than other sports [20,53]. Thus, the first purpose of this study was to assess the body image perception of premenarcheal gymnasts compared to a control group of schoolgirls. It was shown that artistic gymnasts had a positive influence on self-esteem. In particular, gymnasts were associated with a higher body perception compared with sedentary peers. Our data are in agreement with previous studies, which reported a greater body satisfaction in gymnasts than in the general population [54], even in young gymnasts [55]. A previous study has reported that the self-esteem of gymnasts of 6–13 years old, competing at a school level, was significantly lower than that of their peers participating at a recreational level [56]. The discrepancy with our results may be explained because, in their study, a sample of both child and adolescent gymnasts was studied. Thus, the heterogeneity of the age range may have affected the findings of the study. No difference in self-esteem body perception was found in athletes [21]. However, body image dissatisfaction in gymnasts changed over time [57], as former gymnasts had more positive body esteem than current athletes because they suffered more pressure to be thin by their parents. Thus, in order to better understand the issue, it would be interesting to follow a group of young gymnasts in a long-term study to see if changes could occur and to what extent they would have an impact on the self-esteem of body perception, dietary habits and osteoporosis. The latter because it is recognized that physical activity and nutrition influence bone remodeling [58], but there is a lack of information on whether the composition of the meal and the timing of consumption are important for reducing acute bone resorption in children. The results of the study suggest that the consumption of a pre-exercise meal rich in carbohydrates 90 min before the training session attenuates bone resorption in prepubertal female gymnasts with respect to a meal rich in proteins. To our knowledge, this is the first study that investigates the effects of pre-exercise meals on bone resorption in children, and specifically in prepubertal gymnasts. The pronounced bone resorption reduction observed in children following the consumption of the HCM, during the time course, suggests that the manipulation of the nutritional composition of the meal influences bone in the short term. During acute exercise, bone metabolism is unbalanced toward resorption [59]. The marker of bone resorption was measured at four different time points in order to provide an overview of the changes in CTX levels over time. CTX was measured in urine because of the young age of the participants. A fixed quantity of water was provided to the participants to avoid differences in the hydration status and consequently on urine concentration that could impact the study. The effect of meal consumption on bone resorption was analyzed in the morning, after overnight fasting, on the same day and at the same time for all the participants. This was to avoid the day-to-day and within-day variability to which markers of bone turnover are subject to, which may influence interpretation [45]. The gymnasts consumed two different pre-exercise isocaloric meals. In particular, the HCM was rich in carbohydrates. It provided 88% carbohydrates, 9% protein and 3% fat. The HPM provided 55% carbohydrates, 31% protein and 13% fat. The meals were designed by considering recommended values for nutrients for young athletes, considering that children require more energy than adolescents or adults during sports activities [60] and the reproducibility for parents at home. Thus, the consideration to provide a meal easy to prepare, with high palatability for young gymnasts that can be consumed easily by children. It was observed that the consumption of both HCM and HPM before exercise affects acute bone resorption. In fact, in response to feeding, there was an acute drop in resorption, seen as a decrease in CTX levels, in comparison to the fasting state. The ability to reduce the marker of bone resorption was evident with both HCM and HPM and in line with previous studies, in which it has been shown that ingestion of mixed meals, glucose and protein results in a reduction in bone turnover markers [2,45,61]. Thus, regarding the mechanism of action for the observed effects, it is possible to speculate the involvement of the entero-osseous axis, which coordinates bone resorption in response to nutrient intake. In fact, both carbohydrates and protein are able to differentially stimulate the release of the gastrointestinal hormones, such as glucose-dependent insulinotropic polypeptide (GIP) and glucagon-like peptide-2 and 1 (GLP). These gastrointestinal hormones are secreted within a few minutes after the meal [62] and affect acute bone remodeling by significantly reducing bone resorption assessed by the marker of bone resorption CTX [63,64,65,66]. Bone resorption was measured ninety minutes after the meal, and it was more pronounced than that reported in adults in response to oral ingestion and measured for serum and urinary markers of bone resorption [39]. Specifically, in the experimental design it was observed a reduction in urinary CTX by about 77% following the HCM and a reduction by about 62% following the HPM, whereas in adults, it was up to 50% following the meal and glucose and protein consumption [39,45]. We believe that the difference observed in bone resorption in children compared with that previously reported in adults following the meal may be due to the different ages of the participants and suggests that children feeding heavily reduced acute bone resorption. In fact, bone resorption changes as a function of age, gender and puberty but is high in children and adolescents compared with adults [67]. Moreover, it seems to be influenced in the long term by specific activity; for example, by high-intensity physical activity such as rhythmic gymnastics [68]. However, further studies will be necessary before definite conclusions can be drawn. Therefore, whether the consumption of the pre-exercise meal was able to affect bone resorption differently over time was investigated. It was found that HCM, consumed one hundred and eighty minutes before, was still able to decrease CTX levels by about 75% at the end of the training session. Moreover, the HCM after two hundred and forty minutes was still effective in reducing the marker of bone resorption by about 70%. The reduction in CTX levels following the HPM was significantly different and less efficient rather than after the HCM. In particular, the HPM was effective, after ninety minutes of training, in decreasing CTX levels only by about 38% at the end of the workout in the young gymnasts. However, the effect of the HPM, although less pronounced than the HCM, was persistent over time. In fact, the HPM was still effective in reducing bone resorption by about 38% sixty minutes after the conclusion of the training section. These results confirm that the consumption of a pre-exercise meal with high carbohydrates reduced bone resorption in children as previously shown in adult athletes involved in an eight-day overloaded endurance training trial and in individuals during and immediately after a 120 min treadmill run [69,70]. In consideration that bone resorption is highly responsive to nutrient ingestion [64,71,72] and that dietary habits could have an impact on body weight and consequently on body image perception, frequency, the timing of meal consumption and meal type consumed before physical activity were analyzed by using a short FFQ. We found that about 30% of the children do not have a regular pre-exercise meal. In addition, the difference in the timing of meal consumption was wide, ranging from 90 to 30 min before the training section. However, 96% of the girls consumed the pre-exercise meal within this time. Less than 5% of the gymnasts consumed the meal just before physical activity. The pre-exercise meal was for most of the children a fruit juice, fruit or sweet snack. The percent of children that consumed milk or yogurt was low, although at a young age, physical activity and dairy consumption influence bone mass positivity [73]. Overall, the information collected on the eating habits in these groups of children that perform an aesthetic sport suggests that the variability in preworkout meal frequency and timing needs attention to prevent inadequate eating habits. Previous studies on adolescents have demonstrated that sports-involved subjects report greater body image compared to sedentary peers [20,53]. This study, although being conducted on a small sample, may contribute to shed more light on the association between positive body image and sports. The study of the factors influencing the development of a positive body image at a young age is lacking, even though of primary importance. A positive body image is associated with more favorable eating and weight control-related attitudes and behaviors [17], whereas body dissatisfaction is associated with disordered eating and overweight development in future life [74,75,76,77,78,79,80]. Thus, the findings of this study might have practical implications. In fact, by doing sports, specifically artistic gymnastics, children might develop a positive body image. Moreover, artistic gymnastics, by helping to cultivate a more favorable relationship with their bodies in preadolescence, might be used as a tool to prevent poorer psychological health and disordered eating, which are associated with a greater risk of osteoporosis and overweight development. For this reason, cultural and social interventions should be programmed, starting from the school and involving the whole family to promote sports practice along with adequate nutrition. This kind of intervention may be the seed to promote the growth of a healthier population in the near future. 

## 5. Conclusions

In conclusion, this study showed a positive association between participation in aesthetic sports and specifically artistic gymnastic, with components of mental health and emotional well-being and in particular self-esteem of body perception. Furthermore, it was shown that a pre-exercise high-carbohydrate meal is able to reduce significantly bone resorption compared to a high-protein meal in the gymnasts, suggesting that preworkout meal composition should be considered as a factor that preserves bone health in children.

## Figures and Tables

**Figure 1 ijerph-18-02184-f001:**
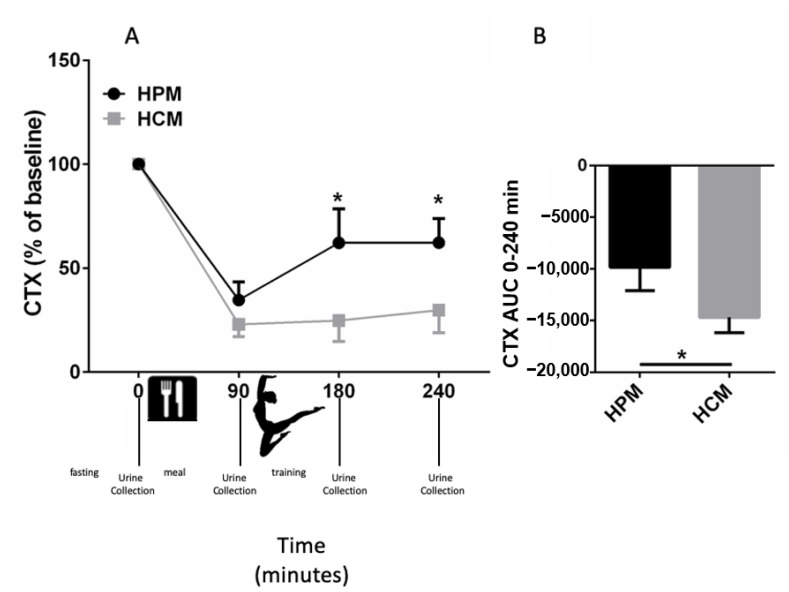
Effects of the pre-exercise meals on collagen type 1 (CTX) levels in female child artistic gymnasts. Urine samples were collected at 0, 90, 180 and 240 min. The meal was consumed after the first urine sample was collected (0 min). The duration of the sports session competition was 90 min and was simulated from time 90 to time 180 min. (**A**) Effects of the HPM and HCM on CTX levels. Mean CTX levels shown as percent of basal level (**B**) Mean area under the curve (AUC) 0 to 240 min of CTX. Error bars depict SEM. Asterisks (*) depict statistically significant differences. Statistical significance is set at *p* < 0.05. *n* = 14 in each group.

**Figure 2 ijerph-18-02184-f002:**
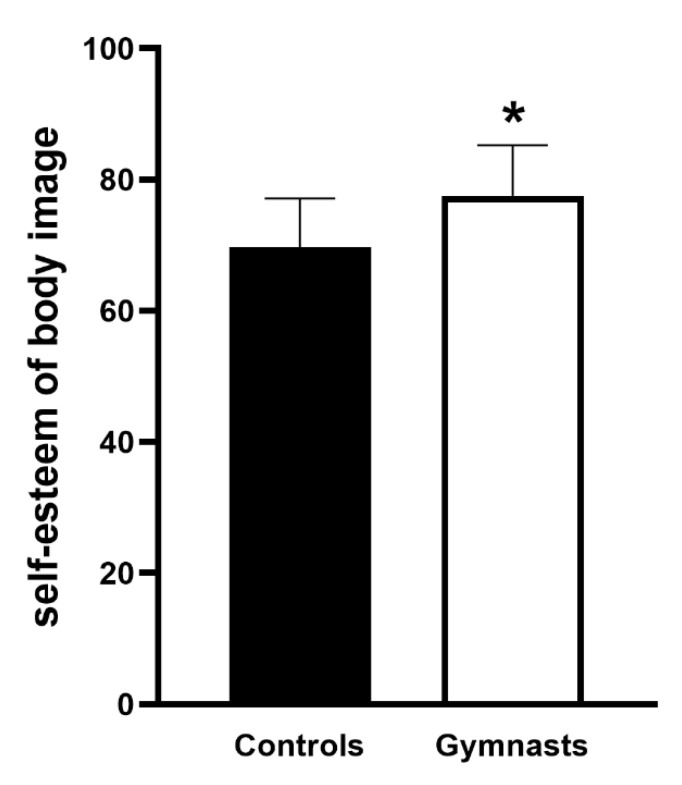
Body image perception of the gymnast group (which included the gymnasts of both the HPM and HCM groups) compared with the control group (the control group consisted of sedentary peers). A subscale of the multidimensional test of self-esteem was used to analyze self-esteem of body image. Error bars depict SEM. Asterisks (*) depict statistically significant differences. Statistical significance is set at *p* < 0.05; *n* = 28 in each group.

**Table 1 ijerph-18-02184-t001:** Energy and nutrient content of the meals.

HPM		**Carbohydrates (g)**	**Proteins (g)**	**Fats (g)**	**Kcal**
	Bread (45 g)	16.5	1.8	0	118
	Baked ham (79 g)	0.3	15.8	7.2	133
	Orange juice (180 mL)	16.3	1.3	0.45	79
	Total	33	19	7.7	330
HCM		**Carbohydrates (g)**	**Proteins (g)**	**Fats (g)**	**Kcal**
	Cereals (60 g)	49	5.5	1.4	242
	Orange juice (200 mL)	18.2	1.4	0.5	88
	Total	67.2	6.9	1.9	330

High-protein meal (HPM); high-carbohydrate meal (HCM); grams (g).

**Table 2 ijerph-18-02184-t002:** Characteristics of the gymnast (both HPM and HCM) and control groups.

Group	Age (Years)		Weight (kg)		Height (cm)		BMI (kg/m^2^)		Fat Mass (%)		Free Fat Mass (%)		Bone Mineral Density (%)	
	Mean	SD	Mean	SD	Mean	SD	Mean	SD	Mean	SD	Mean	SD	Mean	SD
HPM group	10.2	1.2	35. 0	7.2	144	0.1	16.5	1.9	16.4	2.3	83.5	2.3	9.3	0.1
HCM group	10.7	1.0	38.5	9.6	148	0.1	17.2	2.8	17.1	3.0	82.8	3.0	9.4	0.1
Control group	10.62	0.7	39.1	3.6	148.5	1.6	17.9	1.6	-	-	-	-	-	-

High-protein meal (HPM); high-carbohydrate meal (HCM); standard deviation (SD). Values are means ± SD. *n* = 14 in each group.

**Table 3 ijerph-18-02184-t003:** Consumption of the 8 food items included in the short food frequency questions (FFQ).

Food Item	Never		1/Week		2–3/Week		Always	
	*n*	%	*n*	%	*n*	%	*n*	%
Fruit	1	3.6%	3	10.7%	18	64.3%	4	14.3%
Potato chips	2	7.1%	4	14.3%	2	7.1%	1	3.6%
Homemade cake	2	7.1%	1	3.6%	2	7.1%	0	0.0%
Fruit drink	1	3.6%	3	10.7%	20	71.4%	2	7.1%
Milk	2	7.1%	4	14.3%	3	10.7%	2	7.1%
Yogurt	2	7.1%	4	14.3%	4	14.3%	2	7.1%
Savory snacks	2	7.1%	4	14.3%	4	14.3%	2	7.1%
Sweet snacks	1	3.6%	4	14.3%	18	64.3%	3	10.7%

Data are expressed in percent of the frequency consumption of a single item in the week. *n* refers to the number of answers for each item.

## Data Availability

The data presented in this study are available on request from the corresponding author.

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
