# Peer review of "Analysis of Body Perception, Preworkout Meal Habits and Bone Resorption in Child Gymnasts"

_ijerph, 2021, doi:10.3390/ijerph18042184_

Round 1

Reviewer 1 Report

no comments

Author Response

Thank you for the evaluation.

Reviewer 2 Report

Unfortunately, the authors seem not to understand my comments.

My first comment is that the authors claim a cauysal relation between body perception ans gynastics, while I believe there can be an association, but there is not proof for a causal relation. I stonngly suggest to change, in the whole paper, all indications to a causal relation into an association.

Secondly, you measured the effect of a meal on the markers of bone resorption, you can not make any reference as to the excercise, as you did not compare participants having a meal only and participants having a meal and excercise. So, make clear you only studied the effect of a meal, not the effect of exercise.

Author Response

Dear Reviewer,

We have modified the paper as you suggested. We hope that the changes we made on the basis of your suggestions, together with the comments and suggestions of the other referees, and our explanation will satisfactorily answer your questions and remarks for the acceptance of the manuscript.

-My first comment is that the authors claim a causal relation between body perception and gymnastics, while I believe there can be an association, but there is not proof for a causal relation. I strongly suggest to change, in the whole paper, all indications to a causal relation into an association.

Author response: As you suggested we changed the indications to a causal relation into an association between body perception and gymnastics. If there is any missing of the replacing please indicate to us and we will adjust it promptly.

-Secondly, you measured the effect of a meal on the markers of bone resorption, you cannot make any reference as to the exercise, as you did not compare participants having a meal only and participants having a meal and exercise. So, make clear you only studied the effect of a meal, not the effect of exercise.

Author response: As you suggested we have modified the manuscript to clarify better that we studied the effects of the pre-exercise meal and not of the exercise. If there is any missing of the replacing please indicate to us and we will adjust it promptly.

Reviewer 3 Report

The work entitled "Analysis of body perception, pre-workout meal  habits and bone resorption in children gymnasts" has an interesting approach for publication in Int. J. Environ. Res. it shows how society unconsciously prioritizes caring for body image over health from an early age, as well as the need for educational nutritional intervention. But despite being a magnificent job, there are some questions of form that should be taken into account prior to consider this article for publication.
I attach the coments to author with the changes.
•    The manuscript does not specify the objective of the work, which has to be in accordance with its conclusions. Therefore, I consider that they should include it at the end of the introduction, and in the abstrac.
•    It would be helpful to include the data from the control group in Table 2.
•    Figure 1, in isolation, does not provide data that is not recorded in the text. However, it would be of great help associated with figure 3. In this way, we can relate the data to the activity with a naked eye.
•    Several urine samples were collected in a short period of time, During the intervention period, was any other liquid other than orange juice provided to the participants?. this question needs to be clarified and discussed.
•    Regarding the results on the perception of body image,:
      1. Each of the study groups was compared in isolation with the control group? This question needs to be clarified.
       2. It would be interesting to add the previous data in Figure 2.
•    It would be interesting for the authors to include the cultural or social interventions that they would carry out with respect to the results obtained in the discussion.
•    Bibliographic citations should be checked, a large part is missing the pages.

Author Response

Dear Reviewer,

Thank you very much for your nice comments and suggestions on the manuscript. We hope that the following comments and additions will satisfactorily answer your questions and remarks.

The work entitled "Analysis of body perception, pre-workout meal habits and bone resorption in children gymnasts" has an interesting approach for publication in Int. J. Environ. Res. it shows how society unconsciously prioritizes caring for body image over health from an early age, as well as the need for educational nutritional intervention. But despite being a magnificent job, there are some questions of form that should be considered prior to considering this article for publication.
I attach the comments to the author with the changes.

  •    The manuscript does not specify the objective of the work, which has to be in accordance with its conclusions. Therefore, I consider that they should include it at the end of the introduction, and in the abstract.

Author response: As you suggested the objective of the work has been added in the abstract at the end of the introduction (lines:17-18; 81-87).

  •    It would be helpful to include the data from the control group in Table 2.

Author response: As you suggested data from the control group has been added in Table 2.

  •    Figure 1, in isolation, does not provide data that is not recorded in the text. However, it would be of great help associated with figure 3. In this way, we can relate the data to the activity with a naked eye

Author response: Figure 1 and figure 3 were merged into one figure (figure 1).

  •    Several urine samples were collected in a short period of time. During the intervention period, was any other liquid other than orange juice provided to the participants? This question needs to be clarified and discussed.

Author response: the participants were provided with 500 ml of water to consume ad libitum. As you suggested a sentence has been added in the materials and method section and discussed. Lines (115-117; 269-271)

  •    Regarding the results on the perception of body image:
    1. Each of the study groups was compared in isolation with the control group? This question needs to be clarified.

Author response:   For body image perception analysis, “gymnast group” was considered a single group that contains both HPM and HCM gymnast groups. This “gymnast group” was compared with the control group. This information was added in the methods section (lines 139-141).

  1. It would be interesting to add the previous data in Figure 2.

Author response:   As you suggested we have specified the groups of comparison in the legend of figure 2. We did not compare the difference in body image perception between HPM or HCM and control group because the subjects of the HPM and HCM groups were assigned randomly to these groups only for the administration of the different meals and only once. Therefore, the whole groups of gymnasts were considered for the analysis of the body image perception and compared to the control group.

  •    It would be interesting for the authors to include the cultural or social interventions that they would carry out with respect to the results obtained in the discussion.

Authors response: As you suggested we have added in the discussion a sentence to clarify the impact of interventions (Lines: 341-344)

  •    Bibliographic citations should be checked, a large part is missing the pages

Authors response: The bibliographic citations were double checked and corrected

This manuscript is a resubmission of an earlier submission. The following is a list of the peer review reports and author responses from that submission.

Round 1

Reviewer 1 Report

Comment 1: What are the exclusion criteria for the study? 

Comment 2: In line 69 you mentioned that the participants should be clinically healthy. How is that measured? What parameters are used for this?

Comment 3: The effects of HCM and HPM on bone resorption are interesting. Can you shed some light on the reasons as to why this might be happening? 

Author Response

Dear Reviewer,

thank you for your interesting and stimulating comments and suggestions on the manuscript. We hope that the following comments and additions will satisfactorily answer your questions and remarks.

Comment 1: What are the exclusion criteria for the study?

AUTHOR RESPONSE: The exclusion criteria are: 1) chronic illness 2) taking medication 3) taking vitamins or mineral supplements

The sentence has been added to the manuscript (lines 102-103).

Comment 2: In line 69 you mentioned that the participants should be clinically healthy. How is that measured? What parameters are used for this?

AUTHOR RESPONSE: To verify that the participants were clinically healthy the subjects provided a health certificate for competitive sport activity. The certificate is released by the physician following a medical examination, test of urine (urinalyses), electrocardiogram at rest and under stress, spirography (diagnostic test under the Italian law to be able to practice competitive sports activities – Ministerial Decree 18/02/1982). The sentence has been added to the manuscript (lines 98-103).

Comment 3: The effects of HCM and HPM on bone resorption are interesting. Can you shed some light on the reasons as to why this might be happening?

AUTHOR RESPONSE: Bone turnover has a circadian rhythm with higher bone resorption at night and an acute suppression within hours of the ingestion of a meal. This latter effect is due to the action of the entero‐osseous axis which coordinates bone resorption in response to nutrient intake. In fact, ingestion of a meal results in a decrease in bone resorption because macronutrients are able to stimulate, differently, on the basis of meal composition, the release of the gastrointestinal hormones like glucose‐dependent insulinotropic polypeptide (GIP) and glucagon‐like peptide‐2 and 1 (GLP). These gastrointestinal hormones are secreted within a few minutes after the meal and act on bone remodeling. In particular, they act by reducing bone resorption measured by the marker of bone resorption CTX. Thus, it is possible to speculate the involvement of these gut hormones in the observed effects. Sentence (lines 275-281) and references (67-70) have been added to the manuscript in the discussion section.

Reviewer 2 Report

The paper is well written and well founded. It is also well organized and structured and interesting.

Here are the aspects and suggestions of the review of the article:

-The article that is presented deals with the influence between body perception , eating habits before exercise and its repercussion in some aspects (among them self-esteem).

-The manuscript is clearly written in the essay and is interesting and meets the formal characteristics of a research paper . On a methodological level it is correct. The knowledge of the eating habits of athletes for their impact on performance and health is still topical. It is important, and a strength of the manuscript, which includes control group and experimental group, with biomarkers and several measurements. One of the weaknesses of the research is the N it works with (N=28), although the assessment of it is complete.

 Below are some suggestions that should be included in the article:

-While the theoretical justification is adequate, I would suggest that the authors review and include some of the research that has been done on the impact of eating disorders on the sports population. What research has addressed the relationship between body image and self-esteem?

- The working hypotheses would be advisable as they are not explicitly presented in the manuscript.

- Were the hours of training that they performed, and the hours of sport added outside of gymnastics training, taken into account?

- To deepen the discussion on the relationship that these data found could influence the psychological aspects of the gymnasts, highlighting some practical/applied conclusions from the contribution of the article.

Author Response

Dear Reviewer,

thank you for your kind comments and assessment of our study. We hope that the following comments and additions will satisfactorily answer all your questions and remarks:

Comment 1: While the theoretical justification is adequate, I would suggest that the authors review and include some of the research that has been done on the impact of eating disorders on the sports population. What research has addressed the relationship between body image and self-esteem?

AUTHOR RESPONSE: As you suggested we have added in the introduction section of the manuscript, some of the latest research about the eating disorders on the sports population and their relationship with body image and self-esteem (lines 52-74).

Comment 2: The working hypotheses would be advisable as they are not explicitly presented in the manuscript.

AUTHOR RESPONSE: As you suggested we have added the working hypotheses in the introduction section of the manuscript (lines 86-88)

Comment 3: Were the hours of training that they performed, and the hours of sport added outside of gymnastics training, taken into account?

AUTHOR RESPONSE: Yes, we considered the hours of training they performed in the inclusion criteria (lines 92-93). The hours of sport added outside of gymnastics training were also considered, specifically the participants were not involved in other sport or recreational activity (e.g dancing, biking) organized by the local community at the time of the study. To further clarify the point a sentence was added in the methods section (lines 96-97).

Comment 4: To deepen the discussion on the relationship that these data found could influence the psychological aspects of the gymnasts, highlighting some practical/applied conclusions from the contribution of the article.

AUTHOR RESPONSE: as you suggested we have highlighted in the discussion section the positive outcome for the psychological aspects of gymnasts and in particular for the development in young gymnasts of a positive body image and the prevention of eating disorders. This highlights was added in the discussion section (lines 318-330).

Reviewer 3 Report

In this study a number of questions are raised, 1. what is the selfesteem of girls involved in preadolescent aesthitic sports,2. What is the effect of a meal with either a high carbohydrate or protein content on bone resorption and 3. what is the effect of exercise on bone resorption.Although having these questions might be interesting, putting these questions in one paper can cause confusion. Secondly, as will be explained in more detail, is it needed to have appropriate controls for eachr question. 

My major concerns are the following.

  1. Part on body image. A comparison is made between girls participating in aesthetic sports. The results show that the girls invokved in sports have a higher self esteem. The question is if this is related to the sports participation. To be able to participate in these sports, an atlethic body is needed. So, was the self esteem of these girls not already higher before they started with the sport, did they start with the sport because they had a well formed body. How to conclude that the higher self esteem is the result of participating in gymnastic?
  2. Part on bone resorption. You show levels of the bone resorption marker before a meal, after a meal and at two points after the exercise. The mail is either rich in carbohydrate ot protein. You find a decrease of the marker after the meal. The exercise does not seem to have any effect. still, you link your results to the exercise, based on what?
  3. I feel you do not have proper controls, either for the meal or exercise. I feel you need controls with meal but no exercise and a group without a meal but with an exercise to conclude anything about the relation of exercise with CMX.
  4. What might be the explanation for your difference between the CHO and Prot  groups? 

other comments

  1. Why are results expressed as % of baseline? You have longitudinal results within each childer. How do you account for this in your statistics? Why do youi not show the results for all children?
  2. The English language nees -extensive- editing.

Author Response

Dear Reviewer,

thanks for your revision and the comments on the manuscript. We hope that the following comments and additions will satisfactorily answer all your questions and remarks.

Comment 1:

Part on body image. A comparison is made between girls participating in aesthetic sports. The results show that the girls involved in sports have a higher self-esteem. The question is if this is related to the sports participation. To be able to participate in these sports, an athletic body is needed. So, was the self-esteem of these girls not already higher before they started with the sport, did they start with the sport because they had a well-formed body. How to conclude that the higher self-esteem is the result of participating in gymnastic?

AUTHOR RESPONSE: The comparison for the analysis of body perception was made between gymnasts and sedentary peers (lines 105-107). The gymnasts have greater self-esteem compared to sedentary pears. So, we can conclude that participating in aesthetic sports like gymnastics increases self-esteem.

Comment 2:

Part on bone resorption. You show levels of the bone resorption marker before a meal, after a meal and at two points after the exercise. The mail is either rich in carbohydrate or protein. You find a decrease of the marker after the meal. The exercise does not seem to have any effect. still, you link your results to the exercise, based on what?

AUTHOR RESPONSE: We verified if the different composition of the meal was able to attenuate post-exercise bone resorption. The HPM and HCM group performed the same training. Therefore, any difference during and post training on the resorption marker between the two groups is attributable to the different composition of the pre-workout meal.

Comment 3: I feel you do not have proper controls, either for the meal or exercise. I feel you need controls with meal but no exercise and a group without a meal but with an exercise to conclude anything about the relation of exercise with CMX.

AUTHOR RESPONSE: We were looking for the effect of different meal composition on post exercise bone resorption. Thus, we did not include a control group with meal but not exercise or a control group without meal but no exercise because it would provide other information that was not in the aim of the study. However, we thank the Reviewer for the suggestion that paves the way for new studies in the future.

Comment 4: What might be the explanation for your difference between the CHO and protein groups?

AUTHOR RESPONSE: Bone turnover has a circadian rhythm with higher bone resorption at night and an acute suppression within hours of the ingestion of a meal. This latter effect is due to the action of the entero‐osseous axis which coordinates bone resorption in response to nutrient intake. In fact, ingestion of a meal results in a decrease in bone resorption because macronutrients are able to stimulate, on the basis of meal composition, the release of the gastrointestinal hormones like glucose‐dependent insulinotropic polypeptide (GIP) and glucagon‐like peptide‐2 and 1 (GLP). These gastrointestinal hormones are secreted within a few minutes after the meal and act on bone remodeling. In particular, they act by reducing bone resorption measured by the marker of bone resorption CTX. Thus, it is possible to speculate the involvement of the gut hormones in the observed effects. Sentence and references have been added to the manuscript in the discussion section (lines 275-281)

Comment 5: Why are results expressed as % of baseline? You have longitudinal results within each child. How do you account for this in your statistics? Why do you not show the results for all children?

AUTHOR RESPONSE: The results are expressed as percent of baseline because it allows the reader to follow the change in bone resorption over time with respect to the baseline (point zero-start). Likewise, the studies done by J.J. Holst group, a leader in the study of bone metabolism but also by other scientists (e.g Scott, Appl Physiol 110: 423–432, 2011) have shown results expressed in percent of baseline. The differences at time points between groups were analyzed by two-way repeated-measures ANOVA with Sidak test for multiple comparisons as was described in the paragraph for statistical analysis. The results were shown for groups of children. Data are shown as means and SEM. Differences resulting in a P value < 0.05 were considered statistically significant as reported in the paragraph for statistical analysis.

Comment 6: The English language needs -extensive- editing.

The manuscript has been checked and edited for language and form by a mother tongue English teacher. 

Round 2

Reviewer 3 Report

Unfortunately did the authors not really respond to my main concerns of this paper.

My first concern was that the authors conclude that there is a correlation between aerobic gymnastic and self esteem, that this gymnastic increases the self esteem. As said in my first comment, it might wll be that girls with a higher self esteem decided to take part in this gymanstic, so the higher self esteem is not the result of gymanstic.

My second concern related to the correlation made on the effect of exercise on bone resorption. What the authors show is a reduction in bone resorption in relation to a meal. I still do not see the relation with the exercise. 

Why the data are given as percentage of the first measurement is not explained.